# The Importance of the Sun Symbol in the Restoration of Sámi Spiritual Traditions and Healing Practice

**Francis Joy**

The Anthropology Group, The University of Lapland Arctic Center, Pohjoisranta 4, 96100 Rovaniemi, Finland; fjoy@ulapland.fi

**Abstract:** Today, artefacts of the past have immense value for Sámi shamans, artists, and custodians of culture who are reengaging with their spiritual traditions. A cultural revival is taking place through various applications and approaches. Henceforth, there is an ongoing process of creating a restorative framework mainly based on the work of individuals, through which, drum making and decoration, joiking, sacrificial acts, and forms of divination consisting of various sorts of practices are emerging. One of the central symbols that features prominently amongst the Sámi in relation to their prehistoric cosmology and reuse of symbolism in different contexts with regard to spiritual traditions that helps link past with the present is the Sun. Therefore, the purpose of the descriptive analysis in this research paper examines the application of the Sun symbol to new types of drums made by Peter Armstrand who is a Sámi person, for healing and identity building and some of the contexts they appear within. As a method to elaborate on how the past is utilized in the present, the research material constitutes one short case study involving Armstrand who is a Sámi drum maker and likewise, a healer. To help broaden the fieldwork materials collected, I also refer to an old photograph of a drum and its cosmological landscape.

**Keywords:** Sámi shamanism; drums; cosmological landscapes; healing; cultural heritage; art; spirits; sun

---

## 1. Introduction

The emphasis of the study focuses on a contemporary practitioner of shamanism within Sámi culture and the value he relates to the Sun symbol as a mythical figure that is a source of inspiration within a modern practice in connection with healing, ceremony, and ritual. This is analyzed through the art and work of Peter Armstrand who is a Swedish Sámi, from Kiruna, and is the "Vice Chairman of the Kiruna Sámi Association" Joy (2020, p. 1). The material collected in Kiruna, was gathered between 2014 and 2019, from Armstrand, with regard to in what ways the symbol of the Sun is constructed, utilized, and applied in healing and ritualistic practices through a variety of contexts. Correspondence with Armstrand was also established in 2020 for further clarification of data, relating to prior correspondence.

The collection of the material and its analysis have much value with reference to encountering some of the structures related to Sámi religion in contemporary society. Hence, a study of the reuse and application of ancient drum symbolism is a particular area that is currently emerging within Sámi culture. This is in relation to the representation and implementation of cultural heritage with regard to the construction and subsequent decoration and use of contemporary Sámi shaman drums in various contexts by Sámi persons. Drum use is pronounced from Armstrand as an individual Sámi healer in association with healing practices and rituals that point towards a single contribution regarding the restoration of various spiritual traditions and practices and the transmission of culture.

In contemporary society today, study of the long-term perspective of traditional or pre-Christian Sámi religion pertaining to the broad spectrum of practices that are part of it, such as drum use, seems obscure and thus hard to grasp, as do the practices of divination, ecstatic trance-shamanism, and sacrifice, which are all interrelated. This is because, and as stated by Sámi historian Lehtola (2002, p. 28), "the traditions and history of Sámi religious culture are difficult to trace. The religious culture underwent violent changes in connection with Christian missionizing in the 1600s and 1700s. [ . . . ] It is difficult to reconstruct completely the old world-view from sources written by outsiders".

At the same time, it is also important to learn and understand from what has survived with regard to attributes and features of different elements from within traditional Sámi religion. According to Sámi scholar Helander-Renvall (2016, p. 84), "certain aspects of shamanism can be located that are particularly important when talking about relationships. One aspect is to be found among the symbols drawn on traditional Sámi drums. The Sámi shaman drum (*goavddis*) is an expression of the Sámi cosmological, cultural, and spiritual world picture".

It is helpful therefore, at this juncture, to emphasize that approaching the long-term perspectives on Sámi traditional or pre-Christian religion, is an arduous task as a field of study and not without its problems. Difficulties not only include its many manifestations as well as regional variations, but also, in order to fully comprehend what exactly constitutes Sámi religion both past and present, one requires an intimate knowledge of the language and culture from the inside. Thus, for outsiders, or those who do not live within Sámi culture or speak the Sámi language, studying Sámi religion is a monumental task, but not an impossible one.

Despite the cultural fragmentation inflicted upon Sámi society because of colonialism, a combination of the abundance of literature written on Sámi religion by both insiders as well as outsiders, as well as inquiries into the works and practices of Sámi elders, artists, and religious specialists, *noaidi*, who are open to sharing their knowledge, reveal the following. "[ . . . ] There still exists views and practices amongst the Sámi people that show an unbroken series of links with the past, portrayed through landscapes, animals and art" Helander-Renvall (2016, p. 85). Today, these sources and fragments of the past have immense value for Sámi shamans, artists, and custodians of culture who are reengaging with their spiritual traditions.

A cultural revival is taking place through an assortment of approaches and applications of artistic symbolism and figures within various contexts. Henceforth, there is an ongoing process of creating a restorative framework mainly based on the work of individuals, through which, drum making and decoration, *joiking*, sacrificial acts, and forms of divination consisting of various sorts of practices are emerging from within Sámi culture. These provide some insight and understanding into each of the aforementioned spiritual traditions (drum making, sacrifice, *joiking*) and thus, do orient towards the practice of Sámi religion in a contemporary setting through numerous relationships to culture and heritage. Norwegian scholar Trude Fonneland has written about this in a much broader sense regarding neoshamanism at the Sámi shaman festival of Isogaisa, Norway, in her scholarly work: The Festival of Isogaisa: Neoshamanism in New Arenas (Fonneland 2015). In a similar fashion, another Norwegian scholar, Siv Ellen Kraft, has likewise made a valuable contribution on the subject matter noted above, through her scholarly work: Sami Indigenous Spirituality: Religion and Nation Building in Norwegian *Sápmi* (Kraft 2009).

I have chosen an interview with a Sámi person for this research paper firstly, because I myself have been involved in the practice of shamanism for over twenty years and have worked with Peter Armstrand who has made a valuable contribution to this study. In this sense, there was an element of trust regarding Armstrand deciding to share important information, some of which was of a personal nature. The second reason I chose to interview Armstrand is because there is an emerging network concerning the practice of shamanism from within Sámi culture and therefore, this interaction can be seen as both collaboration and participation within this community for the purposes of research development with regard to knowledge sharing and documentation.

As a method for creating a basis and setting for the presentation of Armstrand's contribution to this study, I initially chose to place the focus of this research on the value, visibility, roles, and functions of the Sun as a celestial deity. The foundation of which is presented within various contexts predominantly within Sámi scholarship in order to comprehend its mythical and cultural influence, both past and present, from inside the culture. The reason for doing this is because the history of the Sámi people is an important factor in relation to how myths have been interpreted within various contexts by Sámi *noaidi*. Henceforth, some of these are depicted through art on the ancient drum presented below, as well as scholarly sources and through the contemporary work of Armstrand, all of which are connected to the past. Conversely, I explore how both Sámi history and the Sun as a mythical figure and historical resource, have been drawn upon as a source of inspiration in literature and shamanic practice. In turn, these provide a deeper understanding and examples of the ways in which certain manifestations of the Sun play a central component within the research in connection with healing, ritual, art, and myths (cosmology), in order to demonstrate the Sun's importance in relation to its purpose as a symbol of communication and unity within Sámi religion, spiritual practices, and literature.

## 2. A Focus on Fieldwork and the Interview

It should likewise be noted that I have accompanied Peter Armstrand to the Sámi shaman festival, Isogaisa, which is an annual event in Lavangen, Tromso, Norway, in 2015, 2016, and 2017. The festival is where people from different parts of *Sápmi*—the Sámi homelands areas, as well as areas outside of *Sápmi*, come together and share their beliefs and practices with each other in relation to ceremony, singing, rituals, and healing, which are different applications related to traditional Sámi religion. The festival is a place where Armstrand sells the drums he makes to other Sámi persons as well as non-Sámi. There are also handicrafts that are sold at the event as well as drum building courses that are held occasionally. The ceremonies and practices that take place at the festival are tied to expressions of Sámi pre-Christian religion within a contemporary setting. These can be seen in terms of veneration for, and engagements with, sacred stones, reverence of Sámi spirits and ancestors, drum use, drum circles and journeys, rituals, ceremonies, and healing practices using the drum and *joiking* a form of Sámi singing. Observation of these applications and associations demonstrate how there appears to be "inter-connected features and links, which show an unbroken link with the past" (Helander-Renvall 2016, p. 85) that are visible within a range of relationships, enactments, and procedures at the event.

As a method to elaborate on how the past is utilized in the present, the research material presented below constitutes a short case study involving Peter Armstrand as an artist, drum maker, and healer. The early photographic material was received from Armstrand at his home between August 2014 and November 2019. I then did another interview in March 2020 by Skype in order to expand on earlier correspondence. I wanted to obtain some background information about how he became involved in the practice of shamanism and this is what Joy (2014, pp. 1–2), responded with.

> "I am 49-year-old healer and artist living in Kiruna, Swedish *Sápmi* with my wife Eva and I am a forest Sámi.
>
> My healing abilities started one day when my son's mother complained of pain in her back. I pulled the pain out of her with my hands. During my childhood days, I spent every weekend at my grandparents' home in Lainio; where I experienced them doing healing work on occasions. From this period, the experiences encountered during time at their home led me to undertake training as a healer, in order to become a professional person in my adult life.
>
> The type of spiritual healing I was trained in was Reiki healing for which I became a Master Practitioner. In addition, I have also trained in Inca Tradition from South America in Munai-Ki which means: I Love You, in terms of energy.

With regard to my family background, one of the stories, which have survived about my great grandfather who was called Vaakina-Pekka (1859–1952), and his abilities was published in a book titled: 'Lainio –Our Home Village', on the 650th centenary of the village of Lainio in 1984. The story describes extraordinary events where Vaakina-Pekka reportedly walked over a swollen river without being swept away. Vaakina-Pekka was known for his humility and fear because of his magical knowledge. In the village he was one of the most famous *noids*. The tools he used for his healing work included a snake skin and a stone which was kept in a special box that had never seen the light of day".

As a reflection of what is noted above, Armstrand's explanation of use of New Age healing practices are illustrative of how a Sámi healer has flexibly utilized and combined such practices with traditional Sámi healing practices by merging them together to give Sámi shamanism global recognition, as both Reiki healing and Munai-Ki are universal healing systems, but the Sámi one is concerned with the local, which he emphasized and which his work is grounded in. However, it should be noted that there is an area of ambiguity by contrast to traditional practice of *noaidivuohta* and the use of New Age spiritual practices among some Sámi shamans, as noted at the Sámi shaman festival Isogaisa.

The aim of the research is to document the proposed case study to the extent that it demonstrates the number of ways that Peter Armstrand, as a Sámi person, uses art, healing, and ritual as methods to bring traditional Sámi practices related to spiritual traditions and Sámi religion into a contemporary setting. This is presented through research of Armstrand's work regarding how he uses drum building, decoration, and a series of practices that not only help with rebuilding Sámi culture, but also transmit cultural heritage, cosmology, shamanism, and myths for the purposes of building and maintaining identity and preserving cultural memory through the processes of remembering.

One of the most important contexts outlined within these applications is the collection of data from Armstrand, which conveys how and why cosmological landscapes are painted onto the drums he makes for therapeutic purposes with regard to communicating with Sámi spirits. The research thus aims to elaborate on how these spiritual powers are called upon in order to receive help when administering various forms of spiritual healing. Within these descriptions, I have attempted to demonstrate links with Sámi pre-Christian religion as a method for illustrating connections between the past and present in order to articulate aspects of continuity of tradition and practices; this has not been easy because of the cross-over with New Age practices.

To help broaden the aforementioned aims, I am including three other painted drums made by Armstrand (four in total), which he provides information about concerning the transmission of culture through painted landscapes depicting the Sun as well as Sámi deities. This is for the purposes of providing additional examples of how the symbols of both the Sun and deities are intricately linked to Sámi language, and in what manner they function as systems of communication and in what ways these can vary in their respective landscape settings. Henceforth, his explanations are aimed at communicating religious beliefs that are part of a social institution, which are an intrinsic part of the very fabric of Sámi society.

To further support the contemporary art and drum landscapes made by Armstrand, an illustration of ancient Sámi drums taken from Ernst Manker's esteemed works: *Die Lappische Zaubertrommel: Eine Ethnologische Monographie. 1, Die Trommel als Denkmal Materieller Kultur. [The Lappish Drum: as an Ethnological Monograph, volumes 1 and 2*, (1938 and 1950) is incorporated. This is included because it helps validate the thinking in the mind of the Sámi *noaidi* who painted the cosmological landscapes on the drum in the seventeenth century. In this sense the drum depicts how using ancient art as a system of communication within Sámi society is linked to religious practices and inter-species communication, which are important as expressions of both culture and identity as well as connection to traditions and cultural memory. By the term 'communication' I am referring to prayer, out-of-body travel, and sacrifice directed towards other spiritual dimensions of reality.

I implemented two approaches to the research. The first concerns the application of empathy in the relational approach to the study concerning communication and cooperation with Armstrand, which made the focus meaningful. What this means is that I extensively studied how throughout the course of the sixteenth and seventeenth centuries Sámi history has been predominantly written by priests and missionaries who were scholars, in relation to religious practices. As a consequence, this led to their misrepresentation in order to bring the Sámi under the control of the Christian Churches, which can be seen as a method to destroy indigenous practices and traditions. In addition, information collected from Sámi persons at that time in connection with their religious practices was used against them in court cases in connections with accusations of sorcery; in fact, some Sámi in Norway, Finland, and Sweden were given death-sentences for using the drum. Therefore, it is important to understand the history of the indigenous Sámi and their culture, traditions, and practices[1].

Henceforth, the approach to the study supports building a foundation as a way of minimizing research practices that are prejudiced by giving Armstrand a voice in the research. Presenting photographs of his drums and using reliable sources written from within Sámi scholarship, validate and affirm their contributions to the study and also help demonstrate in what ways tradition as cultural inheritance can be seen as a flexible and constantly changing concept in relation to the transmission of both material and spiritual culture and practices. My reasons for bringing these elements together as someone who practices shamanism is for the purposes of outlining how in doing so "[ . . . ] [my] quest is to be able to hear, feel, understand, and value the stories of [ . . . ] [the research participant] and to convey that felt empathy and understanding back to the client/storyteller/participant. [Furthermore, and] when relevant, the quest also includes conveying that felt understanding to a broader audience" Gair (2011, p. 134). The collection of data helps with understanding how Sámi sacred drums as artifacts, connected to material culture, play a central role and function in the development and formulation of identity in relation to Armstrand's sense of self[2].

I have chosen Vilma Hänninen's narrative approach as a framework for application of the research method regarding the analysis of the materials. In this case, it has value through its implementation, given that it works rather well when combining both scholarly and artistic material together. For instance, we see its "[ . . . ] ability to bring together various disciplines, as well as bridge the gap between science and art" (Hänninen 2004, p. 69). This helps give the Sámi participant a voice and equal representation that is supported with textual data. In essence, the stories and material collected are deeply embedded in both personal and collective narratives and therefore, the narrative approach "[ . . . ] is a primary way of organizing and giving coherence to [ . . . ] experience" (Hänninen 2004, p. 71).

The inner and outer experiences of Armstrand presented below reflect past and present and are therefore indicative of how cosmology, myths, and shamanism function in connection with various types of religious experiences through multiple relationships and reuse of cultural heritage. Therefore, the implementation of the narrative method provides a framework where "the inner narrative can be seen to serve several functions: it makes sense of the past, provides a vision of the future, defines the individual's narrative identity, [and] articulates values [ . . . ]" (Hänninen 2004, p. 74).[3]

---

1   More about this history can be found in the paper (Joy 2018): The Disappearance of the Sacred Swedish Sámi Drum and the Protection of Sámi Cultural Heritage (2018), by Francis Joy.
2   In addition, I was given permission to undertake the interviews by Armstrand, and I sought permission to take photographs of the drums he made and uses for healing and ritual work, as well as the photographs of drums he sent to me.
3   It is likewise necessary to inform the reader that because of different Sámi dialects and languages, there are various spellings of Sámi terms in relation to spirits, the Sun, and landscapes. Examples of these are expressed in quotes by Sámi scholars especially in relation to the data regarding the roles and functions of the Sun within the Sámi culture in Finland and Norway, for instance.

### 3. Examples of Some of the Roles and Functions of the Sun as a Celestial Deity in Sámi Cosmology from within Academic Sources

From observations of early literature written about Sámi cosmology, oral traditions, and myths, the symbol of the Sun is typically portrayed as a positive healing force that brings warmth and makes the grass grow to feed the reindeer. Sacrifices were made to the Sun, as it was worshipped as a major deity who played a central role in fertility rites and healing because of its power and warmth (see for example, Fragments of Lappish Mythology, Lars-Levi Laestadius [1838–1845] (Laestadius 2002, p. 8); also, the work of Johannes Schefferus, The History of Lapland (Schefferus [1674] 1971)). In terms of portraits of the Sun on a *noaidi* drum in the works of Laestadius (p. 8) and its position and status within Sámi cosmology, editor Juha Pentikäinen notes how

"The celestial being is seen as a female deity of the sky and the second class of gods or deities of the sky include *Beiwe* and *Ailekis Olmak. Beiwe*, the Sun is represented by a quadrangle, figure number 4 on the drum. From each angle of the quadrangle runs a line called *Beiwe labtje* (the Sun's reins, i.e., rays). There were four of these reins and they signify the Sun's power of affecting all four directions of the wind". (Jessen 1767) (Pentikäinen 2002, p. 77)

Furthermore, Sámi professor Helander-Renvall (2005, p. 5) writes about in what ways,

"Older written accounts describe the Sun as the mother of all life and living animals. The Sun is always important to the Sámi. [ . . . ] The Sámi ask the Sun to shine: *Beaivváš* would provide light to wanderers in the mountains, to farers at sea, and to herders searching for lost reindeer. The Sun daughter's many names indicate that in most Sami areas the Sun appeared most often in female form".

Another source from Sámi scholar Anna Westman (1997, p. 31) reflects in what way,

"In almost all circumpolar societies, there exists/existed the concept of female goddesses or 'mothers' who regulate fertility and protect family, especially women during pregnancy and children. ( . . . ) *Biejvve* is also part of this complex. She is the burning fire in the sky, the annually recurring force, which in springtime makes the hillsides turn green and ensures there is food for the reindeer. She protects the reindeer calves during spring and sees to it that women get milk from the animals during summer".

Perhaps one of the most interesting and descriptive stories about the mythology concerning the Sun with regard to its value and position on the Sámi drum and reindeer is found in stories related by Sámi scholar of religion Jelena Sergejeva about Sámi folklore tales from amongst the Eastern Sámi on the Kola Peninsula, north-west Russia. Sergejeva (2000, pp. 235–36) contribution reminds us of how the knowledge of the environment has been important for understanding relationships between the Sámi and the relativity of both truth and values attached to it, which can be seen through Armstrand's work today, as well as what is written by the other Sámi scholars presented below.

"The symbol of the Sun is usually thought to be represented on the Sami shaman's drum as a circle (in the north) or as a rhombus, from each corner of which there is a line (common in the south). These four lines, which are like sunbeams, signify the power of the Sun (Kharuzin 1890; Harva 1915, p. 61; Pentikäinen 1995, p. 120). According to many researchers—for instance Kharuzin, who in turn quotes A. Erman-these lines mean the spreading of power in four directions over the earth. A direct translation of the name of these lines, which are known in research literature as *nealja* because *labikje* (old orthography), is four reins of the sun (Kharuzin 1890, p. 143). [ . . . ] It is no mere chance that there is an analogy between the sunbeam, the lines of the sun's power and reins, because, on the one hand, it is natural from the point of view of an ancient reindeer-breeder. On the other hand, this analogy seems to have deeper significance. In the Sami mythology, the sun was connected with fertility.

Reproduction of the reindeer/earlier wild deer and other animals was thought to be closely connected to the warmth and power of the sun. The ancestors of the Kola Sami illustrated the idea of a relationship between the Sun and fertility in ancient cave paintings."

Sámi archaeologist from the Swedish side of *Sápmi*, Inga-Maria Mulk has, likewise, made an interesting contribution to discussions concerning the Sun and its roles and functions on both drums and in rock art. For example, Mulk (2004, p. 50) notes how on some of the south Sámi drums "the cosmic power is represented as the image of the Sun in the drum's center; as *Tjoarvveahkka*, the deity with horns, in the upper world; as *Mattarahkka*'s three daughters in the everyday (middle) world; and as *Jabmeahkka*, deity of the dead, in the underworld" (Manker 1938, 1950; Mulk 1985, 1994).

A further contribution by Mulk accounts the roles, functions, and manifestations of the Sun as documented in early sources described below by Sámi student Nicolaus Lundius and Jacob Fellman, for example, who refers to the following regarding its value, roles, and functions in Sámi myths and cosmology.

"The Sun belongs to the upper world and is another aspect of the Earth Mother figure. In the Sami worldview, the Sun (*Biejvve*) is feminine and, as Mother *Áhkká*, her role is the creation of life. For example, Nicolaus Lundius in the 1670s recorded that for the Sami the Sun is "Mother of all living creatures" (Lundius 1905; Westman 1997). Almost all drums have an image of the sun placed in a center among the heavenly gods and goddesses, of whom *Mattarahkka* is the most important.

*Mattarahkka* belongs to all spheres but was primarily associated with aspects of the upper world—the south, warmth, the source of life. As the primordial, original or first mother, *Mattarahkka* was a deity with multiple qualities (Fellman 1906; Rank 1949, 1955; Bäckman 1982). Together *Mattarahkka* and *Biejvve* were seen as the cosmic force that created life and ended it. These two deities represented the forefathers and foremothers of the Sami people, as well as symbolizing their belief in reincarnation.

Some of *Mattarahkka's* different aspects were represented in the earthly sphere by her three daughters *Sarahkka*, *Juoksahkka* and *Uksahkka*, who are depicted on Sami drums usually standing together in a line [ . . . ]. *Juoksahkka's* symbol is a bow while *Sarahkka* and *Uksahkka* hold staffs with cleft sticks. *Mattarahkka* might also be found in the underworld as *Jabmeahkka*, the goddess of death. Together these deities symbolized the cosmic force that created life and ended it". (Mulk 2004, pp. 54–55)

It seems almost certain the Sun and its solar power as a deity has been a focal point for reverence and worship and expression in its many forms by the Sámi because of its warmth and healing powers, turning darkness to light and cold into warmth. In addition, and according to the edited works of Ralph et al. (1997, p. 6), in their writing about Finnish Sámi poet and artist Nils-Aslak Valkeapää, it is mentioned how "according to one myth, the Sami are the children of the Sun, and the poet honors that myth". This belief is in Valkeapää's written works titled *The Sun my Father—Beaivi, Áhcázan* (1988).

These literature sources are included because they help us to become familiar with interpretations and scholarly-produced learning about Sami knowledge systems, and in a broader sense communicate how Armstrand uses the traditional knowledge he has to reflect Sámi myths, deities, and cosmological landscapes for the purposes of portraying practices and beliefs today that coherently reproduce aspects of Sámi culture in various contemporary settings.

Regarding a further emphasis and explanation of the importance of the Sun as a healing force amongst the Sámi, the focus now turns to use of a healing drum with the Sun at its center belonging to Peter Armstrand, which is pictured below (Figures 1 and 2). The sacred vessel is decorated with *Áhkká* goddesses, namely, *Mádderáhkká, Sáráhkká, Uksáhkká,* and *Jouksáhkká,* and their significance as co-creators regarding the existence of the Sámi people. When asked to explain the significance of both the Sun symbol and *Áhkká* goddesses painted on the drum landscape and use of the instrument, Armstrand answered in the following way.

> "The drum I use for healing is a bowl type drum I made myself, five years ago, from birch burl, and it has a reindeer skin sewn onto it.

> It is painted with acrylic paints on the skin, and inside the drum are Sámi symbols, which have been burned on to it. I also continue to search for old Sámi symbols. When the Sámi spirits give me symbols, I can use them as well for healing and to put inside the drum.

> The Sámi spirits play a very important role in the healing work I do. I call upon them to help provide knowledge, insight and guidance when helping other persons, and also in my development as a healer. During ceremonies, I call in all the *Áhkká* Goddesses who help protect me and the circle and I use *joiking* when I feel the need to.

> The use of the drum is important because the vibrations from the drum go deep into the body and help to release pain, which many people who come for healing, have. For some people, I cannot use the drum because it is too powerful for them, and the healing power raises issues, which they might not be strong enough or willing to face". (Joy 2014, p. 3)

> "The pictures came to my mind about how to decorate this drum head. My guides gave them to me. The Sun is the source or wellspring of life for us Sámi people and it can help restore life to people who are sick. The Sun is the most powerful deity. A strong combination of earth energy from the female *Áhkká* goddesses incorporated with the masculine power of the Sun (water and fire) are a powerful combination used for healing". (Joy 2019, p. 1)

In reflecting on Armstrand's perspective, as stated above, it is important to understand how this makes a significant contribution to better understanding Sámi healing practices, cosmological orientation, and application of traditional knowledge, and how sharing is one of the ways he is helping to maintain, preserve, and sustain this knowledge.

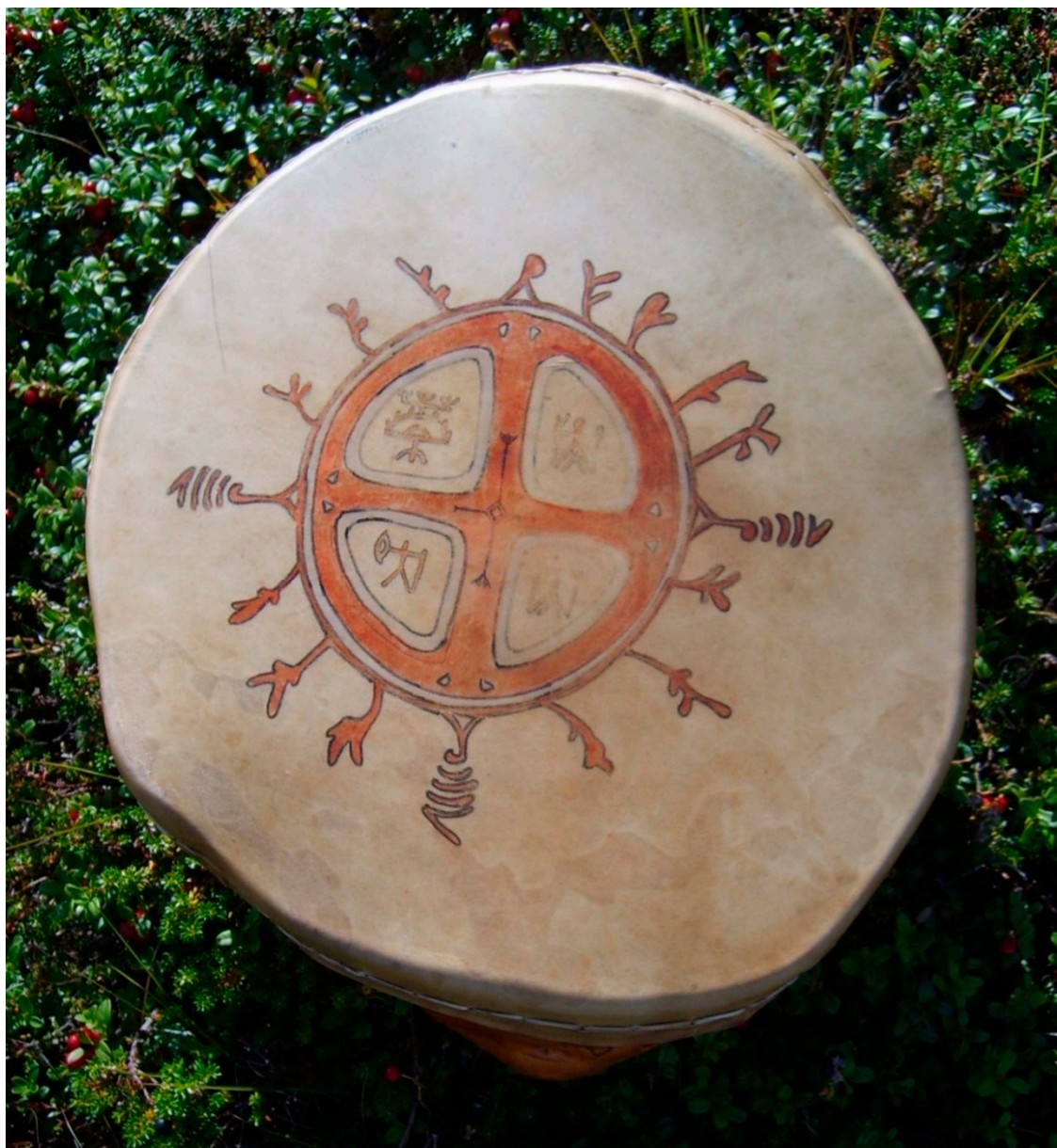

**Figure 1.** The painted Sun symbol on Peter Armstrand's drum, which is divided into four sections that contain illustrations and symbols depicting the Áhkká goddesses. "The Sun's rays are also painted around the outside of the circle as a way to illustrate its healing power" (Joy 2014, p. 8). Photograph and copyright Francis Joy 2014.

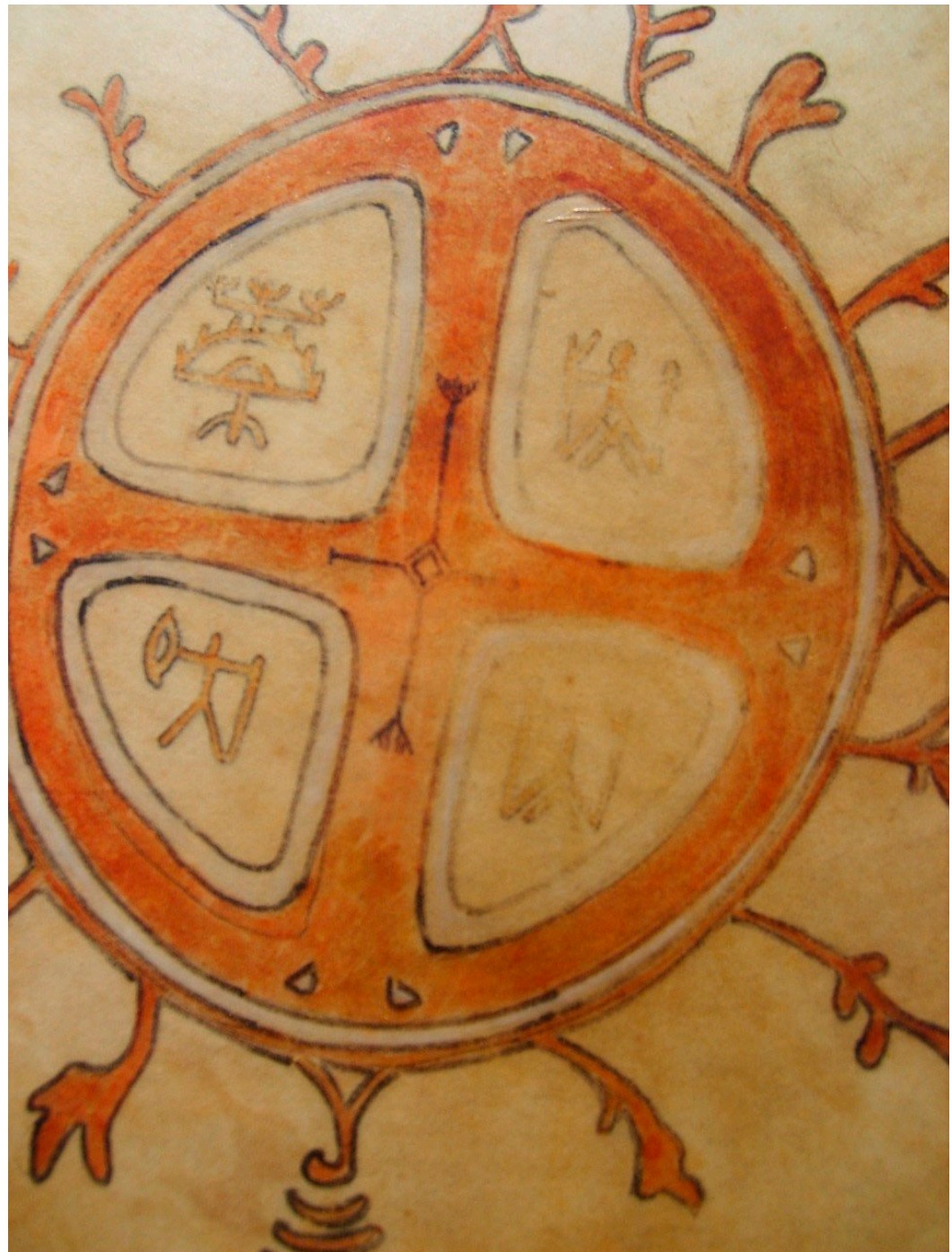

**Figure 2.** A slightly faded illustration of the four female Sámi deities of the Áhkká group are pictures in the Sun symbol, which is divided into the four quarters of north, east, south, and west."*Mádderáhkká*, the Mother Goddess of the Earth is pictured in the top left section. In the top right section is *Uksáhkká*. The bottom left section is *Jouksáhkká*, and *Sáráhkká* is pictured in the bottom right section of the drum" (Joy 2014, p. 10). Photograph and copyright Francis Joy 2014.

## 4. Peter Armstrand's Healing Drum and Interpretations of the Sun Symbol

Observations of the work of Armstrand undertaken during fieldwork helped to understand a number of ways that his behavior as both a drum maker and healer appeared as being contributory in reshaping and rebuilding both individual and collective identity and traditions: linking both past and present together into a unified whole within Armstrand's own personal practices. The purpose of

approaching the subject matter in this manner is firstly because, through this individual case study, evidence was presented on how cultural practices are rebuilt and culture revived and transmitted through art and spiritual practice; where cultural landscapes act as a bridge not only between culture and nature, but past and present as well. Moreover, by formulating the data as it is presented above, this method aims to demonstrate what is presented below in terms of how firstly, tradition is a flexible and constantly changing concept; and secondly, what role the old Sámi cosmological landscapes painted on drums play as elements of material culture, as well as oral traditions and stories in relation to these changes that take place on both social and spiritual levels. Henceforth, linking the past with the present through heritage practices, beliefs, and ancestral memory that are demonstrative of a holistic worldview that has many dimensions to it. Through feedback from Armstrand, in terms of both the healing work administered by him as well as drum making and decoration, it is possible to comprehend in what ways, in connection with his spiritual needs, there is a working relationship and interdependence on transcendent spiritual beings for everyday matters as well as those concerning the development and transmission of culture and tradition.

The *Áhkká* goddesses inside the Sun appear to have two recognizable paradigms to them. The first concerns how Armstrand does not live as a reindeer herder. Armstrand lives in a remote area of a town in northern Sweden and works in a school. Therefore, Armstrand's own cosmology seems very much concerned with the domestic sphere of people and family life. The *Áhkká* goddesses are very much affiliated with the earthly life of family, childbirth, baptism, and human affairs as well as protection of family members and the dwelling place. This might indicate as to why the four goddesses depicted in the drumhead are at the center of his own personal cosmology. In addition to what Armstrand has stated above, I had further personal correspondence with the drum maker on 14 December 2019, concerning the revival of Sámi religion, the important role and function of the Sun as a deity, and why it was important he attended Isogaisa Sámi shaman festival. Armstrand then responded in the following way.

> "For every drum I make and every drum other Sámi drum maker's make, they are bringing back Sámi religion. When I go out into nature and play the drum, I believe the old spirits of nature wake up and want to work with us again. It is slow but the seeds are being sown that will grow again if they are cared for.

> The Sun was one of the highest deities because it made things grow through its light. When the Sun returned after the Polar Nights, people began to celebrate again as it brought new energy. Therefore, the *Áhkká* goddesses on my drum inside the Sun are a symbol of how they and the Sun help the human world with their powers.

> Isogaisa has been a place where we can meet with other people who work in a similar capacity. People from other countries, which has created a unifying experience". (Joy 2019, p. 1)

In relation to Figure 3, the artwork on the drum depicts both old and new symbolism, consisting of a Sámi *noaidi* at the bottom of the drum, holding a drum and hammer and the four *Áhkká* goddesses positioned at each of the quarters of north, east, south, and west.

> "The role and function of *Mádderáhkká*, as the mother goddess of creation stands out because the image of her is larger than the ones of her three daughters. The old Sámi Sun symbol with the new Sámi flag represents the Sámi people and *Sápmi*, the homeland areas. The new image helps illustrate how the Sun still plays a central role and function in cosmology for the Sámi people, giving energy and power to us as well as the whole of Sápmi". (Joy 2014, p. 1)

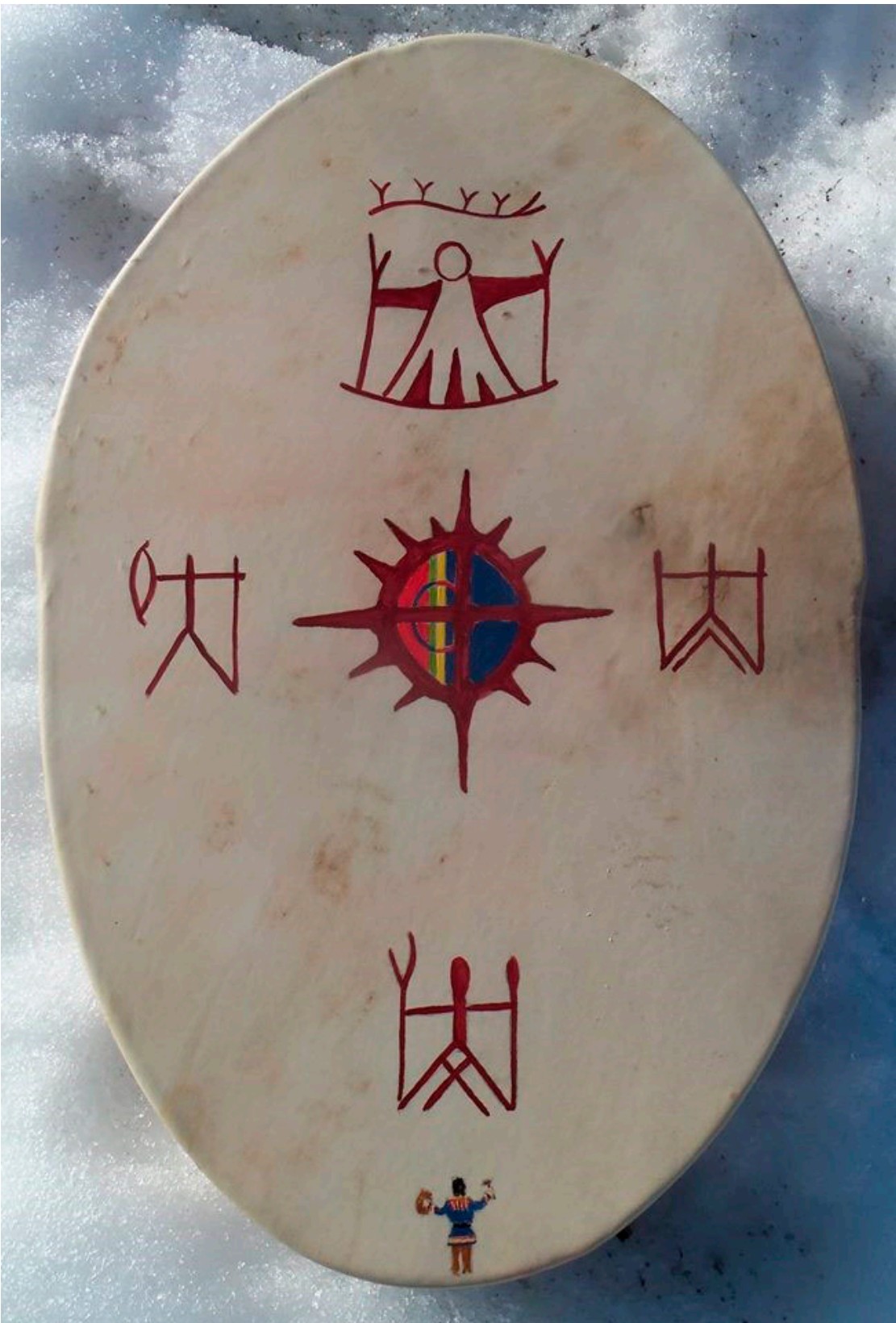

**Figure 3.** A cultural landscape painted on a modern-day Sámi drum also made by Armstrand. Photograph and copyright Peter Joy (2014).

When I later asked Armstrand for a further elaboration of the importance of the Sámi flag inside the Sun symbol, depicted below through figure three, the response was as follows.

"This drum and its content are a representation of Sámi religion. This is because the shaman at the bottom of the drum is drumming for the Sámi people, the Sun and all the Gods. The old symbols and figures are important as they are a source of inspiration for my work. When I make the drums, it is important to make them in a new way but equally as important to keep the old knowledge because it is essential to remember and to reflect on the old customs and traditions.

When I reflect on old drums before making a new one, it is really important to honour and remember the old Sámi *noaidi* who gave their lives for the Christian world. Therefore, in this way, contemporary drum making combines past and present so the cultural memory lives on, as do the religious practices". (Joy 2020, p. 1)

The aforementioned quote by Armstrand is necessary to acknowledge in terms of how the individual reflects on the painful past regarding persecution of *noaidi*, which is how the demise of Sámi pre-Christian religion began. Although, he does not clearly say so, his words give the impression that the events of the past might play a critical function regarding the determination he has for reanimating Sámi traditions and rebuilding culture in a contemporary setting.

With regard to Figure 4, the drums origins are documented as being from "Lule Lappmark" (Manker 1938, p. 791). In the center is a Sun symbol (number 10). What makes this interesting is it is the only Sun symbol on the surviving drums which has a cross inside of it. Therefore, it is possible to understand how Armstrand has sought inspiration from drum number 65 in Manker's book, *Die Lappische Zaubertrommel: Eine Ethnologische Monographie. 2, Die Trommel als Urkunde Geistigen Lebens The Lappish Drum: An Ethnological Monograph. 2. The Drum as a Record of Spiritual Life* (Manker 1950). The Sun symbol in this drum is what motivated Armstrand in his decoration of different drums, the photographs of which the drum maker sent to me, describing the following. "This is a beautiful symbol; small but really powerful. This Sun on the old drum is what inspired me to reuse it on the drums I build" (Joy 2020, p. 1). Examples of these are featured below (Figures 3, 5 and 6).

When I asked Armstrand if it was possible to reflect a little more comprehensively about the shape and design of the Sun symbol on drum number 65 (Figure 4), the drum maker said "after the polar nights are ended in January, there appears an old natural optical phenomenon in the sky when the Sun begins to shine again and there are lots of ice crystals in the atmosphere. The Sun shows himself to us again and quite often through a phenomenon called a Sundog, which is like a large halo around the Sun. The symbol of the Sun on drum number 65 is a good example of this in my mind, and it why I have used it a lot on the drums I make" (Joy 2014, p. 1).

Being able to comprehend this explanation enabled me to further understand the links between the old drum in Figure 4 and in particular, all the contemporary drums made by Armstrand and thus, get a better understanding of the drum makers knowledge, as well as his interpretation of the Sun symbol on Manker's illustration of the drum and its significance.

"[Regarding Figure 5] The drum was made in the winter season and therefore, the image reflects the spirit of this time. The fox is also a representation of freedom and curiosity. The Sun is portrayed with the Sundog phenomena present in its position, partly below the horizon because it only reaches this height during the winter months" (Joy 2014, p. 1). Photograph and copyright Peter Joy (2014). The Sundog phenomena is captured beautifully through Figure 7.

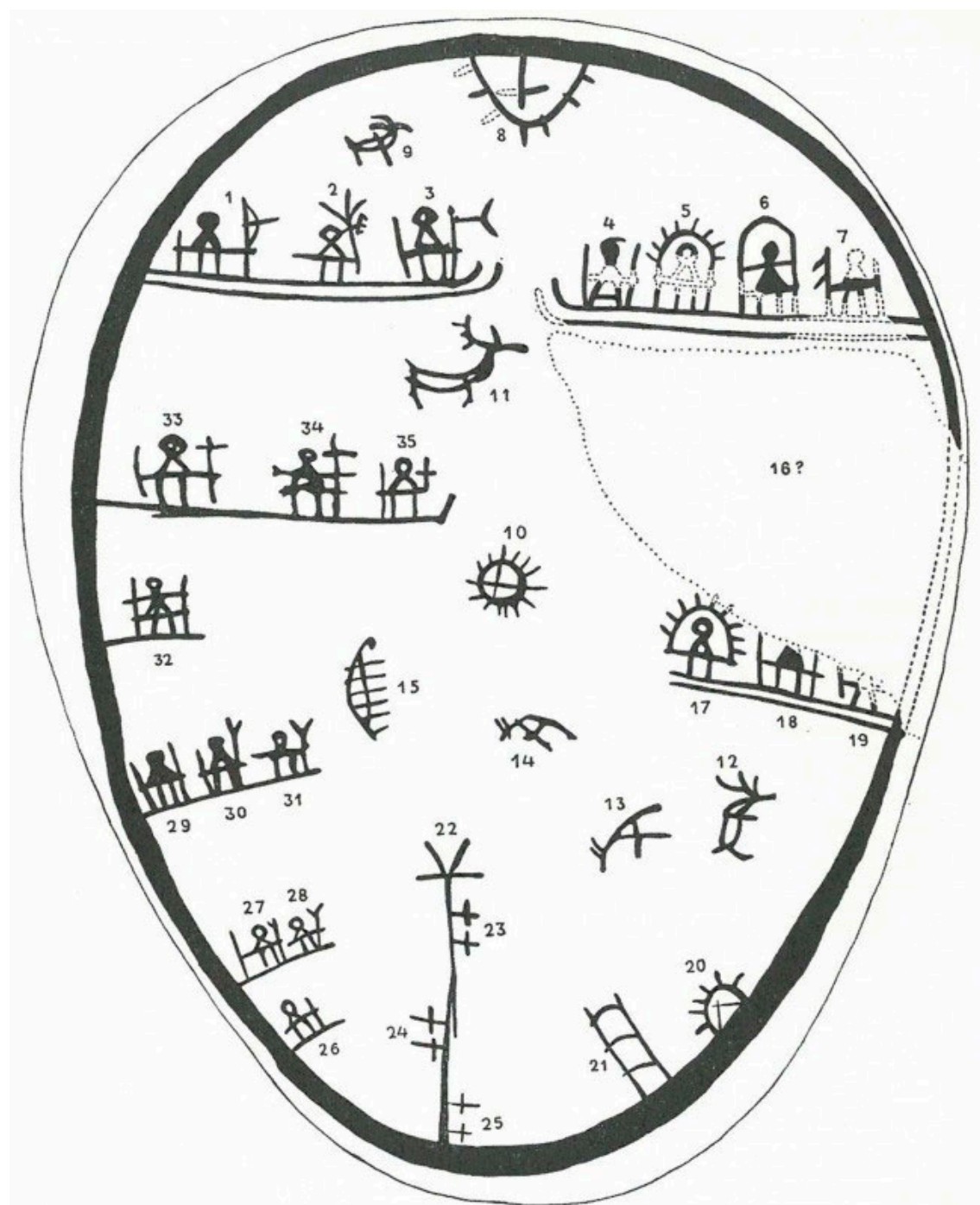

**Figure 4.** A Sámi drum pictured as number 65 in Ernst Manker's inventory (Manker 1950, p. 417).

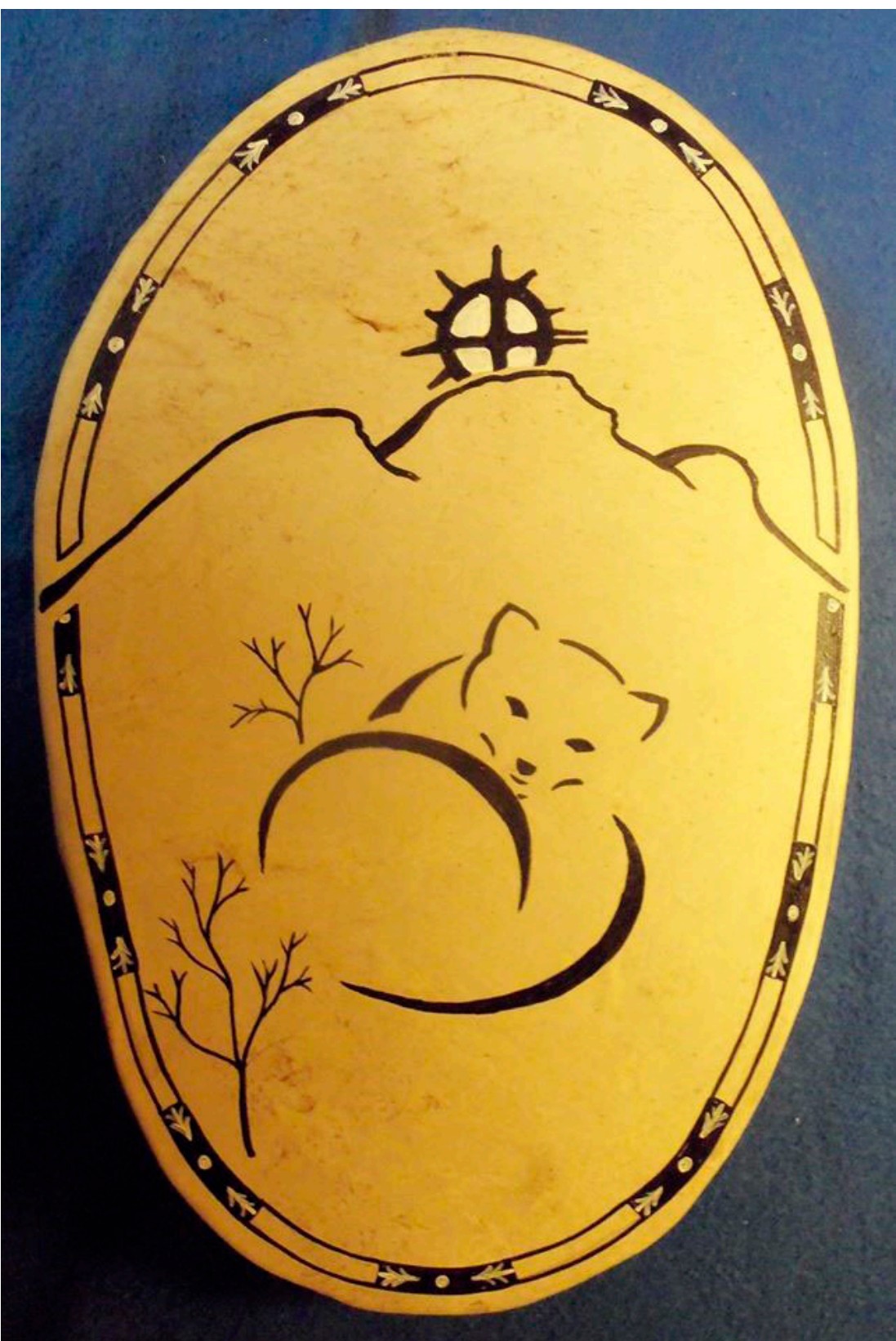

**Figure 5.** An illustration of the spirit of the Arctic Fox. In the background is the Winter Sun and mountains (Joy 2014, p. 1).

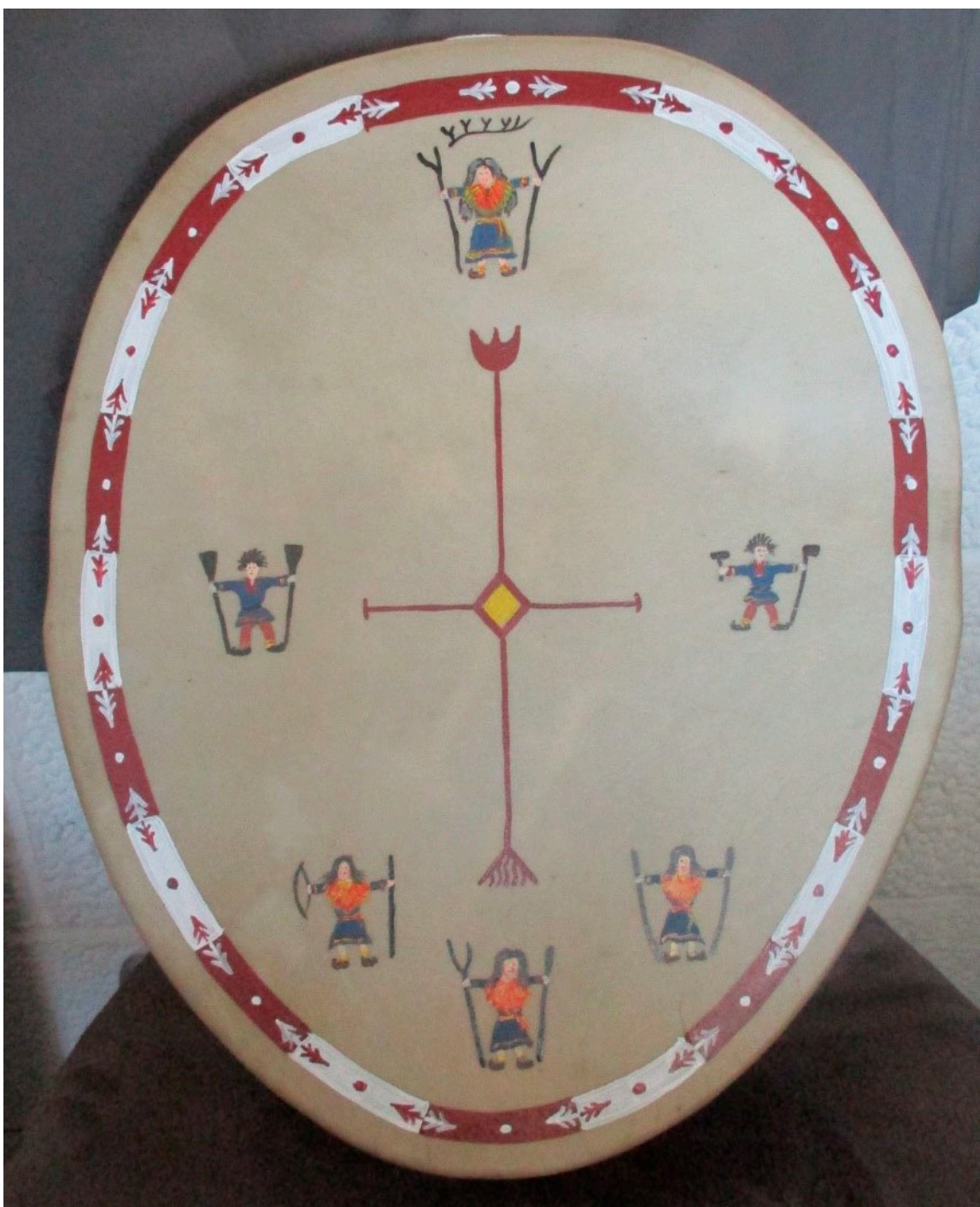

**Figure 6.** Pictured here is a fourth drum made by Armstrand.

"[In relation to Figure 6] the cosmological landscape depicts "*Mádderáhkká*, the Mother Goddess of the earth who is pictured in the top of the drum. In the center of the drum is the Sun and its four rays stretching out in the winter landscape. On the top of the northern ray is a phenomenon that also appears as part of the Sundog or Sun parhelion phenomenon during Winter (as seen on Figure 7). At either side of the Sun rays are the Gods of the wind *Bieggolmái*, and thunder, *Horágálles*. I have characterized human manifestations of the Sámi deities because they are always with us in the human world. In the top right section is the bottom left section is *Jouksáhkká, Sáráhkká and Uksáhkká*" (Joy 2020, p. 1).

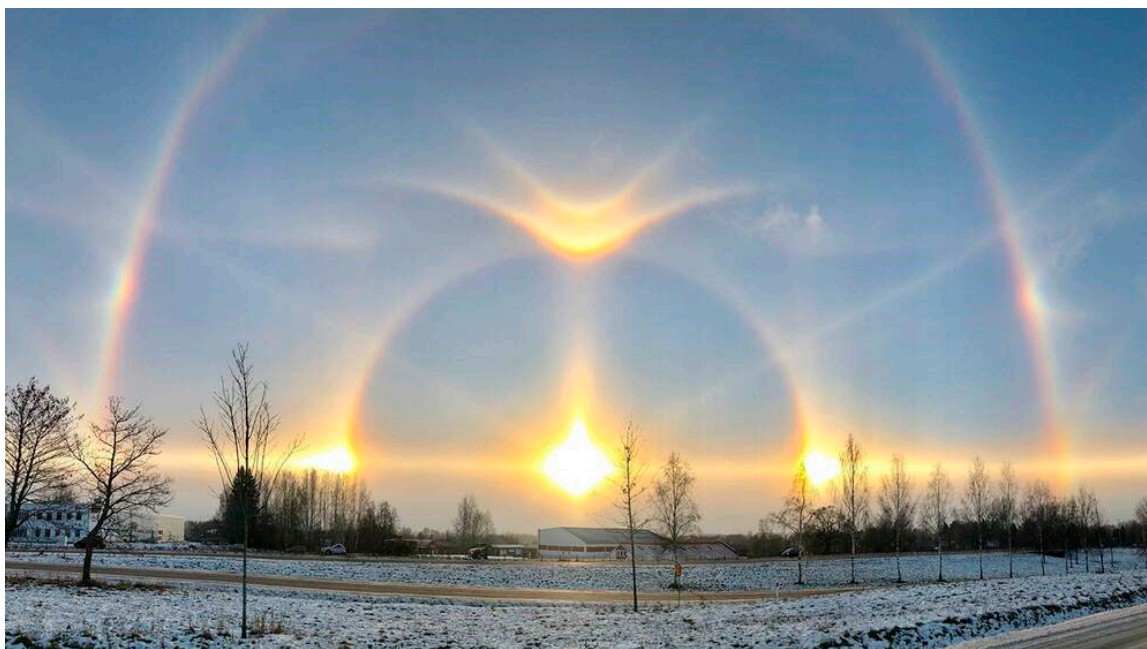

**Figure 7.** An example of the optical phenomena seen here in this photograph. Peter Armstrand wondered whether or not the shape of the Sun at this time of the year was the inspiration for the oval construction of the Sámi drum since time immemorial and thus, one of the reasons why the oval construction was unique to the Sámi people. Photograph taken by Peter Gossas (2018): https://www.svt.se/vader/fragor_och_svar/vad-ar-halo.

## 5. New Perspectives on Sámi Symbolism and Cultural Practices

In terms of trying to establish links between Sámi religion from the past through the contemporary practices documented in the aforementioned with regard to the work of Armstrand, it could be argued how, as part of the processes involved in the construction of identity formulation as well as the transmission of culture and heritage, through drum building and decoration, each of these examples above brings into focus the following points: They provide a broader understanding concerning the important role and function art plays in relation to cultural memory, the creation and preservation of cosmological landscapes on drums, as well as the healing activities drums are used for.

Furthermore, combining art, beliefs, and drum use in relation to their functions as part of the socio-cultural system that underlies Sámi religion, these practices illustrate what Sámi scholar Helander-Renvall (2009, p. 44) refers to as "the inter-subjective character of human and non-human relationships", which are a central component in "Sámi perspectives on the nature of reality" Helander-Renvall (2009, p. 44).

Bringing the stories behind the creation and decoration of new types of drums and their landscapes together, in addition to the literature sources, presents important examples in connection with how the migration and influence of symbolism from the seventeenth century to modern life have taken place through transmission of culture. In this sense, we are able to better understand in what ways the reuse of cultural heritage has a sacred function in creating new knowledge systems that empower Armstrand and the wider Sámi community, thus, adding value and meaning to their implementation and use. The contexts in which beliefs are applied and executed through drum making and decoration, healing practices, and ritual, benefits Sámi culture and reflects Sámi traditional religion because of how Sámi beliefs give "credit [to] natural phenomena [said to be imbued] with spirit and soul, and attributes life to such phenomena as trees, thunder or celestial bodies" (Bird-David [1999] 2002, p. 74).

The explanation of the creation, value, and application of the Sun symbol containing illustrations of the representations of the *Áhkká* goddesses on Armstrand's drums likewise suggests the drums have their own souls and that Armstrand, as the healer, uses the painted drums for establishing

communication with spiritual beings within Sámi cosmology, through prayers and reverence. In this sense, it seems evident he is undertaking the role and function of *noaidi* or shaman as the mediator between the different worlds that comprise the cosmology of the Sámi people as seen manifested through their traditional religion and practices, which are in this case still a living part of daily life.

If we can understand how building and decoration of drums, engaging in healing practices as well as *joiking*, ceremonies and rituals of various kinds are all art forms, then there is a better understanding of the following: The important role and function art plays in the restoration, development, and maintenance of identity and tradition in relation to creating structures and frameworks for drum users, such as Armstrand, to operate within; moreover, where beliefs and perceptions about religion are transmitted. Understanding this helps us see how the past influences the present in connection with transmission of both culture and memory.

Comprehension of Armstrand's work is demonstrative of how both the value and power of ancient symbolism works in relation to healing. Not only because of the Sun symbol and *Áhkká* goddesses depicted on the drum heads, but also the fact that there are ancient symbols inside the drum itself he uses for healing which is evidential of the ways the drum both embeds and embodies the language of the symbols as systems of communication. Henceforth, the language and authority of the *noaidi* is conveyed through drum use and relationships. Moreover, the processes involved in application and use of the drum is symbolic of how Sámi culture is based on symbolism that has been used for a very long time for transmission of culture, identity building, interaction, and interdependency with other life forms from the invisible worlds.

Armstrand's work provides one example of how Sámi knowledge is returning to the world from within the shadows of the past; this is visible to some extent through combining his own life story and that of his ancestors with history and mythology. I would conclude that through Armstrand's approaches to his work, which emphasize systems of communication, the path of healing, and restoration of culture and heritage practices, the interviews and photographic materials presented in this work help to establish, in a number of ways, the means by which Sámi religion plays a central role and function within such processes.

Finally, epic tales and stories perform a critical function in Sámi religion in the way they contain expressions of *Noaidivuohta*, just like stories take up a central position in other religions. The *noaidi* as tradition bearer is one of the key figures responsible for the transmission of experience, expressions, culture, and heritage, through ways of knowing, which takes place when oral memories and myths are transformed into art. Symbolism as such include ecological landscapes, sacred sites, spirits, and departed ancestors; thus, exhibiting the unusual qualities of the *noaidi* as someone who communicates between worlds and remembers.

Inside the middle of the octagon is a central fire representing the Sun. The four lavvus could be viewed as manifestations of the Sun's rays reaching out into the directions of north, east, south, and west because each of the four lavvus' has a fire inside of it.

The symbol of the Sámi flag on the *lavvu* (seen in Figure 8) is also present at the festival each year. According to Kåven and Svonni (2018, p. 1) "the Sami flag represents the Sámi people and *Sápmi* and it carries the Sámi colours red, blue, green and yellow. These colours are most common traditional colours to use on Sámi clothing. It has a circle that symbolizes the Sun in red, and the Moon in blue. The yellow and the green are symbols of nature and the animals".

From observations through attendance at the Isogaisa festival, there is always a fire-keeper present whose responsibility is to keep the fire burning for the duration of the event. Fire, as is seen in Figure 9, plays a central role in Sámi religion in terms of it being a gateway into the spiritual realms for noaidi. Photograph and copyright Francis Joy 2015.

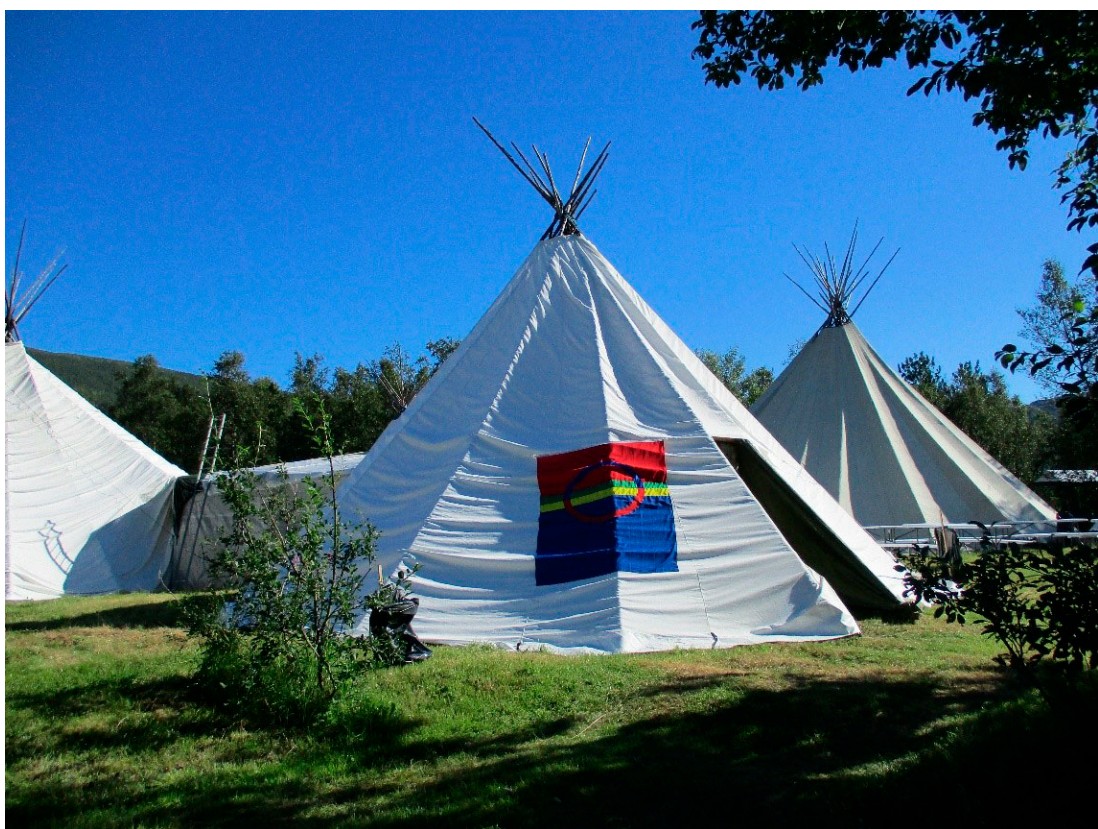

**Figure 8.** As a way of demonstrating further interlocking features and structures associated with Sámi religion and cosmological landscapes, the layout of the ritual landscape at the Sámi shaman festival, Isogaisa, which shows three of the four lavvus connected with the octagon area in-between them, suggests a ritualized landscape. Photograph and copyright Francis Joy 2015.

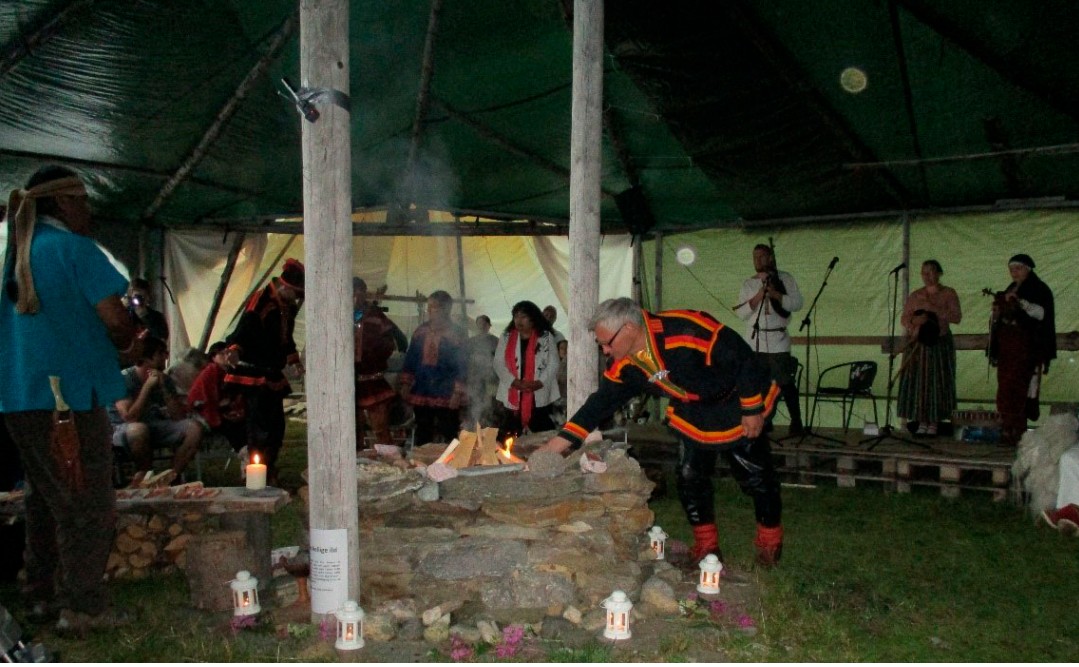

**Figure 9.** Inside the octagon area at the Isogaisa festival where the central fire can be seen here as Sámi shaman Eirik Myrhaug lights it during the opening ceremony in 2015.

## 6. Concluding Remarks

The short case study presented above demonstrates the need for further discussion in relation to formulating a more comprehensive understanding of the relationship between tradition and modernity as a method to help better grasp more broadly what constitutes Sámi religion in contemporary society. Henceforth, a wider understanding is needed regarding how in terms of art, the reuse of ancient symbolism combined with healing practices and ritual act as a bridge between culture and nature that is characterized by relationships. One of the main ways these relationships are portrayed is through painted landscapes, which are applied to the membrane of the Sámi drums which acts as a template for recording religious experiences, reverence, prayer, and remembering. In this sense, the decoration of drums helps create a bridge between the human world and the divine powers in nature and cosmos. Moreover, and in the contexts presented above, how the decoration of the drum is used to form a bridge between the ancient culture of the Sámi and the culture of today.

The construction, decoration, and use of the drums in relation to Peter Armstrand and his work characterizes how for this Sámi person, art and ritual reflects aspects of Sámi culture, religion, and spiritual traditions and practices that emerge from within different contexts, as depicted through the contrasting drum landscapes. This is a way of expressing individual beliefs, thoughts, and feelings, which not only bring healing and empowerment to those seeking help, but likewise, for the participant himself whose determination carries him forward with a sense of pride for being Sámi. Since it has not been possible to draw an in-depth, comprehension of Sámi religion through this one individual case study, Peter Armstrand's contribution to this discussion highlights the need for further dialogue on the subject matter and a much broader study of these ancient traditions and practices, that predate Christianity by thousands of years. This is in order to better understand how traditions are continuously changing within the Sámi worldview and culture is transmitted across generations.

**Funding:** This research received no external funding.

**Acknowledgments:** I would like to express my sincere thanks to Peter Armstrand for his cooperation in writing this paper.

**Conflicts of Interest:** The author declare no conflict of interest.

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
