# Peer review of "The Importance of the Sun Symbol in the Restoration of Sámi Spiritual Traditions and Healing Practice"

_religions, doi:10.3390/rel11060270_

Round 1

Reviewer 1 Report

The paper make an interesting contribution in reporting on a contemporary Sami shaman practitioner, their drum making and the role of the drum and drum making in the Sami revival. I think it needs substantial revision to make it publishable in this journal. Overall is lacks criticality, analysis and engagement with existing literature on Nordic neo-shamanisms. 

The paper needs to cite and discuss literature on insider and empathetic research methods, what is 'tradition' and the role of material culture in identity formation. It needs to engage with this thinking and use it to analyse the case study.

The most promising section is on p.11 when referring to individual and collective identity and tradition, cultural rebuilding and transmission. These sorts of themes need to be at the core of the analysis.  

There are lots of long quotes including from the informant, which are largely left unexamined.

Key sources not cited include Fonneland's 'Contemporary Shamanism in Norway' and Kraft et al's 'Nordic Neoshamanisms'.

The material on the Sun Halo is interesting but a note rather than a key insight, so needs re-framing as such. 

Overall the English language needs improving to make it clearer.

Author Response

Dear reviewer. Thank you for the review of my paper The Importance of the Sun Symbol in the Restoration of Sámi Spiritual Traditions and Healing Practices.

I have made a number of changes to the document and outlined where these are in the text. The most extensive ones being the over-emphasis of the Sundog phenomena, which has been reduced.

In addition, I have also, because of a lack of space, made references to the work of Trude Fonneland and Siv Ellen Kraft and their scholarly works in connection with Nordic Neoshamanisms. 

Again, due to the structure of the article, I have chosen not to engage directly and make comparisons with other shamanic practices in Norway, because the case study has come from Swedish Sápmi and is therefore, somewhat distinct given the fact the subject matter is predominantly concerned with drum making and decoration from this area.

I have also reflected on some of the long quotes, and written my thoughts about these. In addition, I have further more broadly explained-expanded more on page 11, which I am grateful for your comments about, but again this is not extensive because of the limits of the size of the article.

Finally, I have read through the article, which is written in British English, and checked the grammar, spelling and structure and these look fine to me.

Yours Sincerely.

Reviewer 2 Report

This draft article pays testimony to thorough research and experienced academic presentation. It is comparable to the best studies of Tungusic shamanistic healing practices which I am aware of and can therefore be published in the current form.

One query (out of my own ignorance): Are there any studies of the Sámi view(s) of the body, of medicine, of physical healing in pre-modern Sámi society, of travelling healers (other than shamans)? If so, these ought to be referred to and cited.

The language is of sufficiently advanced standard and the article only needs very minor revisions. Examples: one excess 'e' in Eeine  (line 203), rather no semi-colon in line 223, 'sown' and not 'sewn' in line 456, no capital 'I' for 'i' in Russian transliteration ... and why not reproduce the Cyrillic original too? Plus several commas which are misplaced and debatable plural '-s's. Otherwise fine.

An impressive article!

Author Response

Dear Reviewer.

Many thanks for taking time to review my paper: The Importance of the Sun Symbol in the Restoration of Sámi Spiritual Traditions and Healing Practices.

I have addressed the errors you pointed out and thank you for doing so, regarding the apostrophes removed them where necessary, and wanted to say that what you refer to as Russian transliteration is originally English text, so there is no need to add anything more. In addition, I have highlighted in the track changes what has been done regarding amends should you wish to further check them.

Thank you and all the best.

Round 2

Reviewer 1 Report

The revised manuscript is much improved. The discussion of cultural memory, change and transmission is a good addition. There is also a little more reflexivity, which works, although still a lack of attention to methodology (the Gair source is now listed in the references but not cited or discussed). I think the paper could do with a close proof-read from a native English speaker to make some of the phrasing clearer, but as it stands this is acceptable.